**Data Availability Statement:** Data cannot be shared publicly as they are owned by the 45 and Up study. However, anyone wishing to request access can do so through the 45 and Up study.

# The utilisation of public and private health care among Australian women with diabetes: Findings from the 45 and Up Study

**Jon Adams**[1], **Erica McIntyre**[1], **Amie Steel**[1]*, **Brenda Leung**[1,2], **Matthew Leach**[1,3], **David Sibbritt**[1]

**1** School of Public Health, Faculty of Health, University of Technology Sydney, Ultimo, NSW, Australia, **2** Faculty of Health Sciences, University of Lethbridge, Lethbridge, Alberta, Canada, **3** Department of Rural Health, University of South Australia, Whyalla Norrie, SA, Australia

* amie.steel@uts.edu.au

## Abstract

### Aim

To describe the prevalence of health care utilisation and out-of-pocket expenditure associated with the management of diabetes among Australian women aged 45 years and older.

### Design

Cross-sectional survey design.

### Methods

The questionnaire was administered to 392 women (a cohort of the 45 and Up Study) reporting a diagnosis of diabetes between August and November 2016. It asked about the use of conventional medicine, complementary medicine (CM) and self-prescribed treatments for diabetes and associated out-of-pocket spending.

### Results

Most women (88.3%; n = 346) consulted at least one health care practitioner in the previous 12 months for their diabetes; 84.6% (n = 332) consulted a doctor, 44.4% (n = 174) consulted an allied health practitioner, and 20.4% (n = 80) consulted a CM practitioner. On average, the combined annual out-of-pocket health care expenditure was AU$492.6 per woman, which extrapolated to approximately AU$252 million per annum. Of this total figure, approximately AU$70 million was spent on CM per annum.

### Conclusions

Women with diabetes use a diverse range of health services and incur significant out-of-pocket expense to manage their health. The degree to which the health care services women received were coordinated, or addressed their needs and preferences, warrants further exploration. Limitations of this study include the use of self-report and inability to generalise findings to other populations.

Requests can be submitted via email to
45andUp@saxinstitute.org.au for researchers who
meet the criteria for access to confidential data.

**Funding:** The Australian Research Council
(DP140100238) funded the sub-study of women
with diabetes upon which this research paper is
based on and funded Distinguished Professor Jon
Adams via an ARC Professorial Future Fellowship
(FT140100195) while working on this manuscript.
They played no role in the study design, data
collection and analysis, decision to publish or
preparation of the manuscript.

**Competing interests:** The authors have declared
that no competing interests exist.

# Introduction

Diabetes mellitus represents a group of chronic metabolic diseases that are associated with impaired glucose tolerance. An estimated 8.4%(451 million cases) of the global population in 2017 were affected by diabetes(type 1 and type 2) with prevalence rates projected to increase to 9.9% by 2045 [1]. In Australia, diabetes is a national health priority [2], with the prevalence of diabetes in adults aged 20 to 79 years estimated at 6.5%(>1.1 million cases) [3] increasing to 12.3% in women aged 50 years and older [4].

The burden of diabetes for individuals, society and health care systems is high. The global burden of disease study reported in 2013 that diabetes was the seventh leading cause of years lived with disability (YLD) globally, and the tenth in Australia [5]. Contributing to the burden of this disease is the myriad comorbid physical conditions associated with diabetes, including coronary artery disease(CAD), stroke, retinopathy and kidney disease [2]. There is also a high prevalence of depression and distress in people living with diabetes, which can adversely impact a person's level of wellbeing [6]. Poor wellbeing also can be attributed to other factors associated with the disease(e.g. physical impairment, pain), and its ongoing management(e.g. lifestyle adjustment, financial hardship) [6]. Consequently, people living with diabetes tend to have complex chronic health care needs, which require team-based approaches to health care, as well as high levels of health service use [2].

People with diabetes are often required to engage in ongoing self-care practices to effectively manage blood glucose levels, and to reduce the risk of developing diabetes-related complications and comorbidities [2]. Self-care can include regular blood glucose monitoring, foot care, medication use, utilisation of other health care services, and dietary and lifestyle management [7]. Clinical guidelines for the prevention and management of diabetes recommend that evidence-based lifestyle interventions be used to control blood glucose prior to prescribing medications [2]. When a person with diabetes is unable to maintain optimal metabolic control with lifestyle interventions alone, they may be prescribed a range of medications to manage diabetes and related health complications. This can include medications to assist with the management of hyperglycaemia, hyperlipidaemia, hypertension, peripheral vascular disease and mood [2]. Consequently, people living with diabetes can be receiving multiple medications, which require ongoing health monitoring by primary care providers [8].

In addition to standard care, many people with diabetes consult complementary medicine (CM) health care practitioners (e.g., naturopaths, chiropractors, massage therapists) and use CM treatments (e.g., nutritional supplements, herbal medicines, relaxation techniques) to help manage the condition [9–11]. A study from 2010 reported 43% of Australian adults with type 2 diabetes used CM in the previous 12 months [9]; this is higher than that reported in a 2012 US study in which 25% of adults aged 65 years and older with diabetes used CM in the previous year [11]. The Australian study also identified people with diabetes who use CM, compared to non-users, as more likely to have exercised in the previous 2 weeks, and attend exercise or social support groups in the previous 12 months [9]. An additional US study identified participants receiving adjunct naturopathic care for inadequately controlled type 2 diabetes as showing improvements in most patient-reported outcomes, such as improved glucose monitoring, mood and physical activity levels [12]. Meta-analyses of randomised controlled trials have found that the herbal medicines *Trigonella foenum-graecum*(fenugreek) and *Allium sativum*(garlic) significantly reduce fasting blood glucose levels [13, 14].

Poor management of diabetes can result in serious health complications and physical disability, contributing further to the overall burden of the disease [2]. Medication persistence and adherence in type 2 diabetes can reduce unexpected hospital admissions and associated health care costs [8]. However, non-adherence to prescribed treatments for diabetes is

common due to medication side-effects, inconvenience, cost and various other factors [8]. For example, people with diabetes using multiple medications, compared to those who take fewer medications, report poorer quality of life [9]. Due to the complexity of managing diabetes for both patients and practitioners, guidelines also recommend self-management (for type 2 diabetes) as a component of an evidence-based program involving patient education and support tailored to the individual [2]. Poor management of diabetes is a particular concern for women as they are at significantly greater risk of stroke compared to men [15].

The estimated direct health care costs of diabetes in Australia in 2017 was AU$6.9 million (or AU$7,820 per person) [3]. However, the total financial burden of diabetes is likely to be much higher, as this figure does not include indirect costs, such as inability to work and provision of government benefits; it also does not consider the cost of self-care practices or the use of CM, the latter not being considered part of the mainstream health care system in Australia. The economic burden of self-care of chronic illness by individuals and households is often overlooked in Australia in favour of analyses that centre on the macro-economy and the cost to government [16].

Individuals with diabetes often report high utilisation rates of public health care services, particularly women [17]; yet, little is known about the utilisation and cost of accessing private health/self care services in this population. Given the importance of effective diabetes management in preventing potentially serious health complications and comorbidities, which can have a significant impact on financial burden and quality of life, it is critical to understand how people with diabetes utilise all types of health care in order to better inform health services planning and policy development. This is particularly relevant for older women as they are over-represented in the ageing population, experience symptoms of chronic illness differently to men [18] and have specific health needs (often not met by formal expert-led care) [19]. Furthermore, women are known to be greater users of CAM [20]. As such, this project provides an important contribution towards examining the ways in which CAM self-care relates to women's over-representation in chronic illness and their broader disadvantage. Consequently, this study aims to describe the prevalence of health care utilization (including conventional, self-care, and CM) and out-of-pocket expenditure associated with such use for the management of diabetes in Australian women aged 45 years and older.

## Materials and Methods

### Aim

To describe the prevalence of health care utilization (including conventional, self-care, and CM) and out-of-pocket expenditure associated with such use for the management of diabetes in Australian women aged 45 years and older.

### Design

Cross-sectional survey design.

### Participants

The authors obtained data from a sub-study of the Sax Institute's 45 and Up Study, described in detail elsewhere [21]. The baseline questionnaire collected information from 266,848 men and women who reside in the State of New South Wales, Australia and are aged 45 years and above. To establish the 45 and Up Study, individuals meeting these criteria were randomly selected from the Medicare Australia database which provides virtually complete coverage of the general population. Participants entered the study by completing a baseline postal

questionnaire between January 2006 and December 2009 and consenting to have their health followed over time. The study reported here is one of a set of a sub-studies of women from this cohort who had previously indicated that they had been diagnosed with one of five chronic illnesses: asthma, arthritis, depression, diabetes, and osteoporosis. Details of these associated sub studies have been published elsewhere e.g., [22–27]. For this sub-study, the research team mailed a questionnaire to 800 women who reported having diabetes between August and November 2016; 392 returned the questionnaire, representing a response rate of 49.0%.

## Data collection

The questionnaire used for this study included items across four domains–*diabetes status*, *demographic characteristics*, *health care utilisation*, and *out-of-pocket costs*.

**Diabetes status.** The research team asked participants to specify the period (years/months) from when they were first diagnosed with diabetes to the time the survey was administered. Women were also asked to rate the severity of their diabetes during the previous 12 months, on a 10-point scale ranging from 0 (least severe) to 10 (most severe).

**Demographic characteristics.** The research team asked women questions about their demographic characteristics, including date of birth, marital status, ability to manage on their income (i.e., no or little difficulty, some difficulties, struggled), private health insurance status, highest educational qualification, and postcode. Participant postcodes were subsequently converted into Accessibility Remoteness Index of Australia Plus scores in order the define the participant's area of residence (i.e., major city, inner regional area, outer regional or remote/very remote area).

**Health care utilisation.** The research team asked the women to indicate if they consulted any of the following health care practitioners for their diabetes in the previous 12 months: conventional medical practitioners (including general practitioner, medical specialist, hospital doctor); allied health and nursing practitioners (including nurse, pharmacist/chemist, counsellor, psychologist, dietitian, physiotherapist, occupational therapist); and CM practitioners (including acupuncturist, chiropractor, naturopath/herbalist, homeopath, massage therapist, meditation instructor, yoga instructor, nutritionist, osteopath, traditional Chinese medicine practitioner, and 'other' CM practitioner). Participants were also asked by the research team to list any prescription medications and CM products/practices (i.e., aromatherapy oils, herbal medicines, multivitamins, glucosamine/chondroitin, fish oil, homeopathic remedies, meditation without an instructor, yoga without an instructor, physical activities/exercises, and two 'other' CM products/practices options) they had used for their diabetes during the previous 12 months.

**Out-of-pocket costs.** The research team asked participants to report any out-of-pocket costs–defined as direct costs for health services or treatments paid by the user—for consulting a health practitioner, undertaking relevant CM practices, and purchasing relevant CM products or prescription medications for their diabetes over the previous 12-months. These costs were reported in Australian dollars.

## Ethical considerations

The University of NSW Human Research Ethics Committee approved the baseline 45 and Up study and the ancillary study reported on in this paper. The University of Technology Sydney Human Research Ethics Committee also approved the sub-study in accordance with the Declaration of Helsinki. Participants were given a separate information sheet and signed a written consent form.

## Data analyses

A member of the research team used Stata (version 14) to conduct data analysis. The researcher used Spearman's correlation coefficient to examine the association between continuous variables. They also used student's t-test to make comparisons between continuous and categorical variables and chi-square tests to examine the association between categorical variables. The research team extrapolated economic data to the Australian population based on published 2016 Australian population census data. According to the census, in Australia in 2016, 4,165,907 women were aged 50 years and over [28], with an estimated 511,152(12.3%) self-reporting a diagnosis of diabetes [4].

**Validity, reliability, and rigour.** The questionnaire was developed specifically for this project. The manuscript authors along with additional expert advisors produced the draft questionnaire. Pilot testing of the questionnaire was undertaken with lay participants for feedback and face validity and the questionnaire was revised accordingly. The questionnaire is available as a supplementary file.

## Results

### Demographic characteristics

The average age of participating women was 69.6 (SD = 8.7) years, with a minimum age of 53 years and maximum age of 95 years. Almost half of the women (46.0%) resided in a major city, with 40.3% residing in an inner regional area, and 13.7% residing in an outer regional or remote/very remote area. High school education was the highest qualification held by 39.3% of women, followed by a certificate or diploma (28.9%) and a university degree (23.0%), with 8.8% reporting no formal education. The majority of women (60.7%) were married or in a *de facto* relationship, with 32.4% widowed, divorced or separated, and 6.9% single. In terms of ability to manage on available income, 63.1% had no or little difficulty, 23.6% had some difficulty, and 13.3% had struggles. The majority of women (62.6%) had private health insurance.

### Diabetes characteristics

The average period since women had received their first clinical diagnosis of diabetes was 13.6 (SD = 10.7) years. In terms of self-rated severity of diabetes (0–10 scale, with 10 being most severe), the average severity score for women was 3.1 (SD = 2.2) over the past 12 months and 3.0 (SD = 2.2) over the past 4 weeks. Twenty-three percent (n = 90) of women rated the severity of their symptoms as 5 or higher in the previous 12 months. The 12-month and 4-week self-rated severity of diabetes scores were highly correlated ($\rho = 0.90$, $p < 0.001$).

### Consultations with health care practitioners

The majority of women (88.3%; n = 346) consulted at least one health care practitioner in the previous 12 months for their diabetes. Specifically, 84.6% (n = 332) consulted a medical doctor, 44.4% (n = 174) consulted a nurse or allied health practitioner, and 20.4% (n = 80) consulted a CM practitioner in the previous 12 months.

Table 1 shows the average number of participant consultations with each category of health care practitioner, by years since diagnosis of diabetes and severity of diabetes over the past 12 months. Women who reported a diabetes severity rating of 5 or more points (out of 10) had a greater number of consultations with health care practitioners ($p < 0.001$), specifically medical doctors ($p < 0.001$) and nurses/allied health practitioners ($p < 0.001$), compared to women who provided a diabetes severity rating of less than 5 points. Overall, women had attended on

**Table 1. Consultations with health care practitioners by years since diagnosis of diabetes and severity of diabetes over the past 12 months.**

| Diabetes characteristics | | | Average number of consultations | | | |
|---|---|---|---|---|---|---|
| | | | Doctor | Allied health practitioner | CM practitioner | Total |
| | | | Mean (SD) | | | |
| Years since | <10 years | (n = 143) | 3.4 (3.2) | 2.2 (4.0) | 0.7 (1.8) | 6.3 (6.6) |
| diagnosis | ≥10 years | (n = 224) | 3.7 (3.0) | 2.6 (4.3) | 1.2 (3.0) | 7.6 (7.2) |
| | p-value | | 0.323 | 0.279 | 0.107 | 0.095 |
| Severity of | <5 points | (n = 286) | 3.1 (2.7) | 2.0 (3.8) | 1.1 (2.7) | 6.2 (6.5) |
| diabetes † | ≥5 points | (n = 90) | 5.0 (3.5) | 3.9 (5.0) | 0.6 (1.7) | 9.4 (7.8) |
| | p-value | | **<0.001** | **<0.001** | 0.112 | **<0.001** |
| Total sample | | (n = 392) | 3.5 (3.1) | 2.4 (4.2) | 1.0 (2.5) | 6.9 (7.0) |

average 6.9 consultations with health care practitioners for their diabetes in the previous 12 months.

## Use of CM products and practices

Table 2 shows the average number of CM products and practices used by participants, by years since diagnosis of diabetes and severity of diabetes over the past 12 months. There was no statistically significant association between the number of different CM products and practices used for diabetes and severity of diabetes (p = 0.329), or years since diagnosis with diabetes (p = 0.896).

## Out-of-pocket expenses

The out-of-pocket expenses by years since diagnosis of diabetes and severity of diabetes over the past 12 months are presented in Table 3. Women who had been diagnosed with diabetes 10 or more years ago reported greater out-of-pocket expenditure for prescription medications, than women who had been diagnosed with diabetes within the past 10 years (p = 0.042). Women who provided a diabetes severity rating of 5 or more points (out of 10) reported greater out-of-pocket expenditure for CM practitioners (p = 0.039) and prescription medications (p = 0.002), compared to women who provided a diabetes severity rating of less than 5 points.

On average, the annual combined out-of-pocket health care expenditure reported by women with diabetes in this study was AU$492.60 per person. Using 2016 Australian population census data to extrapolate these findings to a national population of women with diabetes, and assuming an average individual out-of-pocket expenditure in line with that of the women

**Table 2. Use of complementary health products and practices by years since diagnosis of diabetes and severity of diabetes over the past 12 months.**

| Diabetes Characteristics | | | Number of complementary health products and practices used | | | | |
|---|---|---|---|---|---|---|---|
| | | | None | 1 | 2 | 3 or more | p-value |
| | | | (n = 229) | (n = 86) | (n = 40) | (n = 37) | |
| | | | % | | | | |
| Years Since | <10 years | (n = 143) | 46.7 | 36.1 | 34.2 | 28.6 | 0.329 |
| Diagnosis | ≥10 years | (n = 224) | 57.3 | 63.9 | 65.8 | 71.4 | |
| Severity of | <5 points | (n = 286) | 76.7 | 74.7 | 79.0 | 72.2 | 0.896 |
| Diabetes † | ≥5 points | (n = 90) | 23.3 | 25.3 | 21.0 | 27.8 | |

† Self-rated severity score out of 10 (1 = least severe and 10 = most severe).

**Table 3. Out-of-pocket expenses by years since diagnosis of diabetes and severity of diabetes over the past 12 months.**

| Diabetes Characteristics | | | Average out-of-pocket expense | | | | |
|---|---|---|---|---|---|---|---|
| | | | Doctor / allied health practitioner | CM practitioner | Prescription medications | Complementary products and practices | Total |
| | | | *Mean (SD)* | *Mean (SD)* | *Mean (SD)* | *Mean (SD)* | *Mean (SD)* |
| Years since diagnosis | <10 years | (n = 143) | $173.4 (284.4) | $46.2 (167.8) | $146.5 (224.1) | $59.4 (185.2) | $425.5 (538.1) |
| | ≥10 years | (n = 224) | $195.9 (325.8) | $81.7 (285.4) | $205.1 (293.4) | $84.6 (233.6) | $567.4 (763.2) |
| | p-value | | 0.498 | 0.179 | **0.042** | 0.277 | 0.054 |
| Severity of diabetes † | <5 points | (n = 286) | $175.2 (294.4) | $76.4 (262.7) | $155.6 (252.5) | $77.4 (225.9) | $484.6 (695.8) |
| | ≥5 points | (n = 90) | $197.8 (328.3) | $18.3 (62.9) | $256.7 (299.4) | $58.3 (124.6) | $531.1 (511.9) |
| | p-value | | 0.537 | 0.039 | **0.002** | 0.444 | 0.558 |
| Total Sample | | (n = 392) | $178.2 (303.0) | $63.9 (239.2) | $175.9 (264.0) | $74.6 (211.6) | $492.6 (671.8) |

† Self-rated severity score out of 10 (1 = least severe and 10 = most severe).

from this study (AU$492.60), we estimate the total out-of-pocket expenditure associated with diabetes treatment for Australian women aged 50 years and over to be approximately AU$252 million per annum. Of this total figure, approximately AU$70 million per annum was spent on CM (practitioners: AU$32 million per annum; products and practices: AU$38 million).

## Discussion

The results of this study provide important insights regarding public and private health care utilisation by middle-aged and older Australian women living with diabetes. The type and range of health practitioners that women are consulting for their diabetes seems to support current practice guidelines. For instance, medical doctors were the most commonly consulted health practitioner by women with diabetes. This finding aligns with current guidelines, which position general practitioners as central to the long-term management and oversight of patients with diabetes [2]. Current practice guidelines also emphasise the importance of self-care and lifestyle interventions for the management of diabetes, and the need for multi-disciplinary team-based care to help address these recommendations [29]. Given the modest prevalence of use of both allied health practitioners and CM practitioners by participants in this study, it is possible that many women with diabetes may be receiving multi-disciplinary care. Notwithstanding, we are unable to determine whether the care received by women with diabetes aligns with patient preferences [30] or is coordinated across all health services. Closer examination of these issues is therefore warranted.

Our analysis suggests approximately AU252 million is spent out-of-pocket for diabetes management by Australian women aged 50 years and over. The individual cost calculated from our study (AU492.60) figure is significantly lower than the AU7,820 per person aged 20–79 years as estimated in 2017 by the International Diabetes Federation (IDF) [3]; however, the expense estimates provided by the IDF encompass a wide range of ages, health services and costs incurred at all levels of government beyond those borne by individuals and households out-of-pocket. Equally, it is not clear whether these previous estimates include out-of-pocket expenses for individuals with diabetes, particularly in relation to self-care and CM health services, as these generally sit outside the conventional, government-funded health system in

Australia [31]. It is important that future estimates of total health service expenditure explicitly account for the full range of health services and treatments by individuals with diabetes.

There are a known number of cost-saving interventions for diabetes, including pharmaco-therapy(e.g. the use of ACE inhibitors for hypertension, angiotensin receptor blockers or early irbersartan treatment for the prevention of end-stage renal disease), and multi-component treatments for diabetic risk factor control and early detection of both type 1 and type 2 diabetes [32]. Intensive lifestyle coaching, health behaviour counselling and screening interventions were also found to be cost-effective. However, effective implementation of these types of interventions is dependent on patients having adequate health literacy, self-efficacy and problem-solving skills [33]. Many primary care practitioners may not have the time or training required to provide the intensive support needed by patients to effectively implement and sustain life-style change [34]; although, emerging evidence does suggest CM practitioners, such as naturo-paths, may be able to fill this health service gap [12, 35]. Given that some CM practices have been shown to be cost-effective in managing other chronic diseases [36], further interrogation of the economic impact of utilising diverse health services to manage diabetes is needed. Also, in light of the differences in out-of-pocket spending patterns observed (in our analysis) between women with varying levels of symptom severity, future economic research should give separate attention to the impact of using multiple health services for individuals experiencing diabetes symptoms of low and high severity.

An interesting observation in our study was that women diagnosed with diabetes 10 or more years prior to completing the questionnaire (compared to less than 10 years) had higher out-of-pocket expenses for prescription medications. This finding is important as it highlights that a longer duration of diabetes is associated with greater risk of comorbidities and, subse-quently, a greater real or perceived need for treatments. Even though increasing diabetes dura-tion is associated with higher risk of comorbidity [37] (and greater actual/perceived need for prescribed medication), effective management of diabetes arguably extends beyond pharma-ceutical treatment. In fact, diabetes management requires long-term coordinated team-based communication between all types of health care providers, patients and health care systems [2]. Future research should examine the challenges of such care co-ordination and the impact of co-morbidities on health service use in older women with diabetes.

## Limitations

This is the first known study to examine the use of a broad range of health services and prac-tices by Australian women aged 45 years and older and living with diabetes. The study presents findings from an established, internationally recognised cohort study. However, our findings need to be considered within the context of its limitations. First, the study did not distinguish between type 1 and type 2 diabetes. There are both similarities and differences in the treatment and management needs of type 1 and 2 diabetes, and previous studies have identified the diffi-culties in distinguishing between the two types of diabetes in population studies using similar recruitment methods [38]. Our research is a retrospective cross-sectional study using self-report; as such, our data is subject to recall bias. The risk of such recall bias was reduced with the use of established survey methodology including measures previously employed in similar cohorts [39–41].

## Conclusions

The findings of this study suggest older Australian women may use a range of conventional, nursing/allied health, CM and self-care interventions/services to manage their diabetes and pay a considerable amount out-of-pocket to access this care, The economic burden placed on

these women warrants further investigation to understand how health care services (and the integration of such services) can better address the biopsychosocial needs of this population. Examining how multi-disciplinary, patient-centred and evidence-based health care services can be effectively delivered to women living with diabetes will assist with the translation of these findings into clinical guidelines and care.

## Acknowledgments

This research was completed using data collected through the 45 and Up Study (www.saxinstitute.org.au). The 45 and Up Study is managed by the Sax Institute in collaboration with major partner Cancer Council NSW; and partners: the National Heart Foundation of Australia (NSW Division); NSW Ministry of Health; NSW Government Family & Community Services—Ageing, Carers and the Disability Council NSW; and the Australian Red Cross Blood Service. We thank the many thousands of people participating in the 45 and Up Study.

## Author Contributions

**Conceptualization:** Jon Adams, David Sibbritt.

**Data curation:** David Sibbritt.

**Formal analysis:** David Sibbritt.

**Funding acquisition:** Jon Adams, David Sibbritt.

**Methodology:** Jon Adams, Erica McIntyre, David Sibbritt.

**Project administration:** Erica McIntyre, Amie Steel.

**Supervision:** Jon Adams, David Sibbritt.

**Writing – original draft:** Jon Adams, Erica McIntyre, Amie Steel, Brenda Leung, Matthew Leach.

**Writing – review & editing:** Jon Adams, Erica McIntyre, Amie Steel, Brenda Leung, Matthew Leach, David Sibbritt.

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
