## [Decision Letter · Decision Letter 0]

1 Mar 2021

PONE-D-20-37288

The utilisation of public and private health care among Australian women with diabetes: Findings from the 45 and Up Study

PLOS ONE

Dear Dr. Steel,

Thank you for submitting your manuscript to PLOS ONE. After careful consideration, we feel that it has merit but does not fully meet PLOS ONE’s publication criteria as it currently stands. Therefore, we invite you to submit a revised version of the manuscript that addresses the points raised during the review process.

We look forward to receiving your revised manuscript.

Kind regards,

Vijayaprakash Suppiah, PhD

Academic Editor

PLOS ONE

Journal Requirements:

2. Thank you for submitting the above manuscript to PLOS ONE. During our internal evaluation of the manuscript, we found significant text overlap between your submission and the following previously published works.

- https://doi.org/10.1080/02770903.2020.1741609

We would like to make you aware that copying extracts from previous publications, especially outside the methods section, word-for-word is unacceptable, even for works which you authored. In addition, the reproduction of text from published reports has implications for the copyright that may apply to the publications.

Please revise the manuscript to rephrase the duplicated text, cite your sources, and provide details as to how the current manuscript advances on previous work. Please note that further consideration is dependent on the submission of a manuscript that addresses these concerns about the overlap in text with published work.

3. Please include additional information regarding the survey or questionnaire used in the study and ensure that you have provided sufficient details that others could replicate the analyses.

For instance, if you developed a questionnaire as part of this study and it is not under a copyright more restrictive than CC-BY, please include a copy, in both the original language and English, as Supporting Information. Moreover, please include more details on how the questionnaire was pre-tested, and whether it was validated.

4. Please provide additional details regarding participant consent, specifically for the sub-study described in the manuscript .

In the ethics statement in the Methods and online submission information, please ensure that you have specified (i) whether consent was informed and (ii) what type you obtained (for instance, written or verbal).

If your study included minors, state whether you obtained consent from parents or guardians.

If the need for consent was waived by the ethics committee, please include this information.

6. Please amend the manuscript submission data (via Edit Submission) to include author Erica McIntyre.

Reviewers' comments:

Reviewer's Responses to Questions

**Comments to the Author**

1. Is the manuscript technically sound, and do the data support the conclusions?

Reviewer #1: Yes

Reviewer #2: Yes

2. Has the statistical analysis been performed appropriately and rigorously? 

Reviewer #1: Yes

Reviewer #2: Yes

3. Have the authors made all data underlying the findings in their manuscript fully available?

Reviewer #1: Yes

Reviewer #2: Yes

4. Is the manuscript presented in an intelligible fashion and written in standard English?

Reviewer #1: Yes

Reviewer #2: Yes

5. Review Comments to the Author

Reviewer #1: The article is Accepted in the present form. This study was done on women and it is important to know the use of various medical services and their expenditure on management of diabetes. It would have been better if type 1 and type 2 diabetes data was separately presented.

Reviewer #2: In the current manuscript, authors attempted to understand the prevalence of health care utilisation and out-of-pocket expenditure associated with the management of diabetes among Australian women aged 45 years and older.

The authors investigation results, may help to frame modifiable trajectories to prevent comorbidities associated with diabetes:

I have few queries and suggestions related to the above manuscript:

• Authors were not highlighted, why it is important in the women population

• Definition of Out of pocket expenses need to be mentioned briefly

• Authors were difference between the type 1 diabetes and type 2 diabetes. Further

• In table they can bold the significant value for better visibility

• Table 3: $46.2 (167.8), 167.8 ? need to be indicated in table

• Investigator results shows that “older Australian women may use a range of conventional, nursing/allied health, CM and self-care interventions/services to manage their diabetes, and pay a considerable amount out-of-pocket to access this care. One of the important aspect in this is older women will have co-complications history among diabetes subjects, Hence I suggest authors to consider the co-complications in to the analysis.

6. PLOS authors have the option to publish the peer review history of their article (what does this mean?). If published, this will include your full peer review and any attached files.

Reviewer #1: No

Reviewer #2: No

---

## [Author Response · Author response to Decision Letter 0]

7 Jun 2021

Response to reviewers is attached as a separate document

---

## [Decision Letter · Decision Letter 1]

21 Jul 2021

The utilisation of public and private health care among Australian women with diabetes: Findings from the 45 and Up Study

PONE-D-20-37288R1

Dear Dr. Steel,

We’re pleased to inform you that your manuscript has been judged scientifically suitable for publication and will be formally accepted for publication once it meets all outstanding technical requirements.

Kind regards,

Vijayaprakash Suppiah, PhD

Academic Editor

PLOS ONE

Reviewers' comments:

Reviewer's Responses to Questions

**Comments to the Author**

1. If the authors have adequately addressed your comments raised in a previous round of review and you feel that this manuscript is now acceptable for publication, you may indicate that here to bypass the “Comments to the Author” section, enter your conflict of interest statement in the “Confidential to Editor” section, and submit your "Accept" recommendation.

Reviewer #1: All comments have been addressed

Reviewer #2: All comments have been addressed

2. Is the manuscript technically sound, and do the data support the conclusions?

Reviewer #1: Yes

Reviewer #2: Yes

3. Has the statistical analysis been performed appropriately and rigorously? 

Reviewer #1: Yes

Reviewer #2: Yes

4. Have the authors made all data underlying the findings in their manuscript fully available?

Reviewer #1: Yes

Reviewer #2: Yes

5. Is the manuscript presented in an intelligible fashion and written in standard English?

Reviewer #1: No

Reviewer #2: Yes

6. Review Comments to the Author

Reviewer #1: The submitted article in revised form is Accepted for being published in the journal in my opinion.

Reviewer #2: Authors modified the manuscript according to the reviewer comments.

I am satisficed with the revision, Hence my recommendation is to accept to publish.

7. PLOS authors have the option to publish the peer review history of their article (what does this mean?). If published, this will include your full peer review and any attached files.

Reviewer #1: No

Reviewer #2: No

---

## [Editor Report · Acceptance letter]

3 Aug 2021

PONE-D-20-37288R1 

The utilisation of public and private health care among Australian women with diabetes: Findings from the 45 and Up Study

Dear Dr. Steel:

I'm pleased to inform you that your manuscript has been deemed suitable for publication in PLOS ONE. Congratulations! Your manuscript is now with our production department. 

Kind regards, 

on behalf of

Dr. Vijayaprakash Suppiah 

Academic Editor

PLOS ONE